# Demographic and Clinical Characteristics of Mpox Patients Attending an STD Clinic in Lisbon

**DOI:** 10.3390/ijerph20196803

**Published:** 2023-09-22

**Authors:** Margarida Brito Caldeira, José Miguel Neves, Mafalda Pestana, Rita Corte-Real, Maria José Borrego, Rita Cordeiro, Jorge Machado, Ana Pelerito, Isabel Lopes De Carvalho, Maria Sofia Núncio, Cândida Fernandes

**Affiliations:** 1Dermatology and Venereology Department, Centro Hospitalar Universitário de Lisboa Central, Entidade Pública Empresarial, 1169-050 Lisbon, Portugal; 2Molecular Biology Department, Centro Hospitalar Universitário de Lisboa Central, Entidade Pública Empresarial, 1169-050 Lisbon, Portugal; 3Infectious Diseases Department, National Institute of Health Dr. Ricardo Jorge, 1649-016 Lisbon, Portugal

**Keywords:** monkeypox virus, sexually transmissible disease, outbreak, human-to-human transmission

## Abstract

Mpox is a viral disease caused by the monkeypox virus, which marked the year of 2022 with a global outbreak. While previously considered to be a zoonosis of almost exclusive animal-to-human transmission, the current outbreak has been attributed to human-to-human transmission, particularly sexual transmission. As a new sexually transmissible disease, we studied the epidemiological and clinical features, as well as the concomitant occurrence of other sexually transmissible diseases, treatment approach, and outcome of our 291 patients, in the current outbreak. We found a total of 169 concomitant sexually transmissible infections of bacterial and viral origins, corresponding to 107 patients. *Neisseria gonorrhoeae* was the most common agent, particularly in the anal location. With this work, we emphasize the need for a thorough epidemiological and medical history, as well as a concomitant complete laboratorial screening for other STIs in patients with confirmed or suspected mpox.

## 1. Introduction

In 2022, the world faced a global outbreak of a previously rare zoonotic disease caused by the monkeypox virus, a member of the *Poxviridae* family and *Orthopox* genus, which also includes the smallpox virus [1]. Formerly known as monkeypox infection, the World Health Organization recommended “mpox” as a preferred term since November 2022, in order to reduce stigma towards the disease, while the virus designation remains the same [2]. Mpox is a zoonosis endemic to central and western Africa, with animal-to-human transmission from its natural reservoir (small African rodents) via bites or direct contact. Subsequent human-to-human transmission is possible, either through respiratory droplets, fomites, or direct contact with a skin or mucous lesion of an infected individual [1]. After the eradication of smallpox and discontinuation of routine smallpox vaccination, the level of cross-immunity to mpox decreased, thus creating a younger generation more susceptible to this disease [3]. Three distinct viral clades have been identified thus far: clade I (formerly known as the Central Africa clade) and clades IIa and IIb (both previously known as the West Africa clade) [4,5]. The former is associated with more frequent and severe complications, and thus a worse prognosis than the latter [6]. 

Since the beginning of this epidemic, more than 80,000 confirmed cases of mpox have been reported in 111 countries [7]. A total of 130 deaths worldwide were attributed to mpox, a fact consistent with the identification of clade II, which has a better prognosis [7]. In this outbreak, the first confirmed mpox case was identified in the United Kingdom, and notification of the WHO occurred on the 13 May 2022 [8]. Subsequently, Portugal and Spain identified the first cases on 18 May 2022 [9]. Global dissemination of the virus reached worldwide proportions, with cases all over the world.

Given the exponential number of new confirmed cases since May 2022, human-to-human transmission of monkeypox virus, which was previously considered to be limited, was found to be the main form of transmission in this outbreak. The infection occurred predominantly in males, particularly men who have sex with men (MSM), with skin lesions primarily found in the anogenital area [1]. Surprisingly, transmission during sexual intercourse was found to be the main form of acquisition of the infection. High viral loads are found in the urine, saliva, semen, feces, oropharynx, and rectum, thus explaining the sexually transmissible character of the infection [10]. Concomitant bacterial and other viral sexually transmissible agents are found in a variable percentage of mpox patients, a fact that corroborates mpox as a sexually transmissible disease [11].

The virus can penetrate from any skin or mucous surface, replicating at the inoculation site, with subsequent spread to regional lymph nodes and viremia. Incubation period ranges from 7 to 21 days. Prodromal symptoms result from viremia and consist of fever, lymphadenopathies, headache, myalgias, and malaise, with possible contagiosity at this time. After 1 to 2 days, skin and mucous lesions begin to develop, initially at the inoculation site, with a variable degree of dissemination [1]. The total number of lesions can vary from a few to hundreds. Within the next 2 to 4 weeks, skin and mucous lesions evolve synchronously from erythematous macules to papules and pseudo-pustules. The latter may simulate pustules, as the content appears to be purulent, but are in fact papules predominantly filled with cellular debris, and no pus or fluid is found [12]. Pseudo-pustules then develop a central umbilication and a deep-seated appearance, covered by a crust that spontaneously detaches 7 to 14 days after lesions begin. From this point, patients are no longer considered contagious. Regarding diagnosis, swab of a cutaneous lesion and posterior real-time PCR assay is considered the gold standard for detection of the monkeypox virus [12].

At present, there are no established clinical treatments for mpox infection that have been scientifically proven [13]. Similar to many viral illnesses, the approach to treatment focuses on managing symptoms, providing supportive care, and isolation measures. Antiviral therapy, including tecovirimat and brincidofovir, which are approved for the treatment of smallpox, is used in selected patients [14]. Indications include severe disease (e.g., involvement of multiple organ systems, hemorrhagic disease, encephalitis), engagement of anatomical regions that can lead to significant complications (such as scarring), immunocompromised patients (e.g., advanced HIV-1 infection, leukemia, lymphoma), children (particularly if <1 year of age), pregnant or breastfeeding woman, and patients with a dermatological disease where skin barrier function is impaired (e.g., atopic dermatitis). Tecovirimat is an oral intracellular viral release inhibitor, restraining viral envelope protein VP37, and thus blocking viral maturation as well as the release of the virus from infected cells. The prescribed dosage of tecovirimat is determined by the patient’s weight. For instance, individuals weighing between 40 kg and less than 120 kg are advised to take 600 mg every 12 h. The duration of therapy is usually 14 days. Brincidofovir, the prodrug of cidofovir, is an oral DNA polymerase inhibitor, with in vitro activity against the monkeypox virus. Oral suspension and tablets are available, with dosage dependent on the patients wight, given at days 1 and 8. Vaccinia Immune Globulin (VIG) is licensed for the treatment of complications of vaccinia vaccination, and it is not approved for the treatment of mpox. However, it can be considered for prophylactic use in an exposed person with severe immunodeficiency in T-cell function for which smallpox or mpox vaccination following exposure to mpox virus is contraindicated [13,14].

Vaccination against mpox is achieved using a highly attenuated, nonreplicating form of the vaccinia virus (Modified Vaccinia Ankara), an *orthopox* which belongs to the same genus as mpox and smallpox. Indications for vaccination include pre- and post-exposure prophylaxis. Pre-exposure prophylaxis should be considered in people with a high risk of acquiring the disease, such as patients with a diagnosis of a sexually transmitted infection in the past six months, sex workers, and healthcare personnel. Post-exposure prophylaxis should be considered in patients who had a high-risk exposure to mpox (e.g., sexual contact with a confirmed mpox patient), with vaccination within 4 days of the exposure [13,14].

There were 953 confirmed cases in Portugal, of which 291 were identified and evaluated in Centro Hospitalar Universitário de Lisboa Central. Herein, we describe the epidemiology, clinical features, concomitant diseases, treatment approach, and outcome of our patients, in the current outbreak. We emphasize the need for a holistic approach to these patients, including a complete sexually transmissible infection (STI) laboratory screening upon first medical visit, with subsequent treatment when applicable.

## 2. Materials and Methods

The patients with laboratorial confirmation of infection with the monkeypox virus (via real-time PCR) observed in Centro Hospitalar Universitário de Lisboa Central in 2022 were included in this study. In total, 291 cases were identified.

Demographic information such as gender, age, and nationality was obtained for every patient. Furthermore, an epidemiological questionnaire regarding previous travels, contact with unknown people or travelers, history of unprotected sexual intercourse, close contact with a symptomatic individual, and attendance of parties or events with multiple people was made. A detailed sexual and medical history was obtained when suitable concerning sexual orientation, past and present illnesses, current medication, known allergies to drugs, presence of systemic symptoms (such as fever, muscle pain, arthralgia, weight loss, night sweats, pain), location and evolution of skin and mucous lesions, and presence of lymphadenopathies.

Samples for monkeypox virus sequencing were collected: one swab at the base of a skin or mucous lesion, one in the oropharynx, one in the rectum, and a blood sample. These were sent to the Portuguese National Institute of Health (Instituto Nacional de Saúde Doutor Ricardo Jorge, INSA, IP). A complete sexually transmitted disease screening was also obtained from 230 patients, including blood samples to test for HIV, HCV, and syphilis (using VDRL and TPHA); urine sample, oropharynx, and rectal swabs were collected to test for *Neisseria gonorrhoeae* (NG) and *Chlamydia trachomatis* (CT) using the Nucleic Acid Amplification Test. The collected samples were screened at the CHULC laboratory for CT/NG via real-time PCR using Cobas^®^ 4800 CT/NG (RocheDiagnostics, Amadora, Portugal).

All 291 patients tested positive for monkeypox virus DNA via the real-time PCR technique. These data were retrospectively collected, consulting patients’ medical information.

## 3. Results

### 3.1. Demographic Information (Table 1)

A total of 290 patients were male, with only 1 case identified in a female. The mean age was 34 years, ranging from 19 to 63 years of age. Overall, 127 patients were of Portuguese nationality, while the remaining 164 were foreigners.

**Table 1 ijerph-20-06803-t001:** Epidemiological, sexual, and clinical data.

Data		*n* (%)
Sex	Male	290 (99.7%)
	Female	1 (0.3%)
Sexual orientation	MSM	281 (96.6%)
	MSW	5 (1.7%)
	MSWM	4 (1.4%)
	WSM	1 (0.3%)
Medical history	Known HIV infection	119 (40.9%)
	Under PrEP	79 (27.1%)
Location of cutaneous lesions *	Genital area	136 (46.7%)
	Anal and perianal areas	116 (39.9%)
	Upper limbs	76 (26.1%)
	Face	59 (20.2%)
	Trunk	42 (14.4%)
	Oro-labial mucosa	30 (10.3%)
	Average (range)	1.61 (1–6)
Lymphadenopathies	Inguinal	130 (44.7%)
	Cervical	21 (7.2%)
	Axillary	11 3.8%)

MSM: men who have sex with men; MSW: men who have sex with women; MSWM: men who have sex with women and men; WSM: women who have sex with men; PrEP: pre-exposure prophylaxis for HIV. * cumulative.

### 3.2. Epidemiological Data

In the previous month, 15 patients reported traveling to various destinations: 4 to Spain, 4 to England, 3 to Germany, 3 to Belgium, and 1 to Brazil. During the same period, 75 patients admitted to engaging in unprotected sexual intercourse with unfamiliar individuals, while 42 patients disclosed having unprotected sexual intercourse with partners who had confirmed mpox infection. Furthermore, 40 patients attended parties or crowded places where sexual activities could be facilitated. The remaining 119 patients did not respond to the epidemiologic questionnaire or did not identify any possible situation where transmission could have occurred.

### 3.3. Sexual History and Clinical Data (Table 1)

Overall, 281 patients identified as MSM, while 5 identified as men who have sex with women (MSW), 4 as men who have sex with men and women (MSMW), and 1 patient identified as a woman who has sex with men (WSM).

Regarding previous medical history, known infection with HIV was found in 119 mpox-confirmed patients, while 18 patients had a history of cured HCV infection. Most patients with HIV infection were under regular treatment, with undetectable viral load in 108 patients. Concerning the 172 HIV-negative patients, we found that 79 were under daily pre-exposure prophylaxis (PrEP) for HIV.

Mild fever (under 38.5 °C) was the most common systemic symptom, affecting 177 mpox patients. Localized mild-to-moderate pain, most commonly affecting the anogenital area, was found in 156 patients, with severe anal pain reported by 9 patients.

Cutaneous lesions affected an average of two body locations, ranging from one to six simultaneously affected sites. The genital area was the most commonly involved (affected in 136 patients), followed by the anal and perianal areas (116 patients), upper limbs, face, trunk, and oro-labial mucosa. Cutaneous and mucous lesions presented as whitish papules or pseudo-pustules which became centrally umbilicated and covered in a hematic crust, with a spontaneous detachment with time.

Superficial lymphadenopathies were found in 162 patients with confirmed mpox. The most common location was the inguinal area, followed by the cervical and axillary regions. These lymphadenopathies were mildly painful upon palpation, of elastic consistency, and mobile.

### 3.4. Outcome, Complications, and Treatment

Most patients had mild disease, with a scarce number of lesions and a good general state. Ambulatory care consisting of oral pain medication and antipyretics was prescribed with favorable outcomes. Patients were advised to avoid direct contact with other people and to abstain from sexual activity until all cutaneous and/or mucous lesions had healed, which happened on average after 21 days.

Regarding complications, one patient was admitted to the hospital due to bacterial superinfection of an mpox lesion on the face. Treatment with oral antibiotic was sufficient to achieve cure without sequelae. Severe anal pain similar to acute proctitis was reported by nine patients, especially when mpox lesions occurred in the anal and perianal areas, with accompanying pain with defecation. Most of these cases were treated in ambulatory care, with regular medical visits, and treatment consisted of optimized pain medication and topical anesthetics such as lidocaine gel, in order to reduce discomfort, with resolution of these symptoms in about 3–5 days. Two cases of patients with severe anal pain were admitted due to failure of pain control in outpatient care, with prompt recovery. No deaths due to mpox were registered.

### 3.5. Sexually Transmitted Diseases Screening Results (Table 2)

A total of 169 concomitant sexually transmitted diseases were found at the time of mpox diagnosis in a total of 107 patients. This indicates that 46.52% of confirmed mpox patients that undertook the STI screening had at least one concomitant STI at the time of diagnosis. On average, each positive patient had 1.58 concomitant sexually transmitted diseases, ranging from 1 to 4 simultaneous infections. *Neisseria gonorrhoeae* was the most common agent, in a total of 61 identified infections: 33 in the ano-rectal location, 24 in the oropharynx, and 4 in the urine. A total of 58 cases of concomitant *Chlamydia trachomatis* infection were identified: 37 in the ano-rectum, 8 in the oropharynx, and 13 in the urine. Early syphilis was the third most common concurrent infection, with a total of 44 cases, most of which were in the form of secondary or early latent syphilis. Inaugural infections with HIV and HCV were identified in four and two patients, respectively. All patients with identified concomitant infections were treated accordingly or referred for appropriate medical counseling.

**Table 2 ijerph-20-06803-t002:** Concomitant infections of the study population.

Data		*n* (%)
*N. gonorrhoeae*	Ano-rectal location	33 (19.5%)
	Oropharynx	24 (14.2%)
	Urine	4 (2.4%)
	Total	61 (36.1%)
*C. trachomatis*	Ano-rectal location	37 (21.9%)
	Oropharynx	8 (4.7%)
	Urine	13 (7.7%)
	Total	58 (34.3%)
Syphilis		44 (26.0%)
HIV (newly diagnosed)		4 (2.4%)
HCV (newly diagnosed)		2 (1.2%)
Total		169 (100%)

HIV: human immunodeficiency virus; HCV: hepatitis C virus.

## 4. Discussion

The year of 2022 was marked by the global emergence of a previously rare zoonotic disease caused by the monkeypox virus. This smallpox-related virus, belonging to the *Orthopox* genus, gained attention due to its rapid spread and significant impact on public health. Although previously considered to be mainly transmitted via animal-to-human contact, the current worldwide outbreak of mpox cannot be solely explained by this form of transmission. Human-to-human contact, particularly during sexual contact, was the main form of dissemination of the disease [15]. The public health impact of mpox during the 2022 global outbreak has raised considerable concern. The widespread transmission of mpox across different countries and continents has presented a significant challenge for global health systems. The ease of human-to-human transmission, particularly through sexual contact, has facilitated the rapid spread of the virus beyond its usual endemic regions. The large number of mpox cases has put a strain on healthcare systems in affected regions. Increased hospitalizations, treatment demands, and the need for isolation and infection control measures have challenged medical facilities and resources. The unexpected revelation of mpox as an infection that can be transmitted through sexual contact has raised awareness about the risks involved in sexual activities, especially among MSM. It has highlighted the importance of sexual health education and preventive measures among vulnerable populations.

Demographic data from our study are consistent with other mpox works, as the majority of confirmed mpox patients were male (*n* = 290, 99.7%), of young age, and most identified as MSM (*n* = 281, 96.6%). Epidemiological information allowed the tracking of a few possible links between confirmed cases. Although no patient traveled to an mpox-endemic country, traveling history could explain a probable method of global dissemination of the disease, as most patients with recent travels reported to have engaged in sexual activities with unknown partners at the location of the travel. Recent unprotected sexual intercourse was found in 75 patients, demonstrating a risk factor for transmitting and acquiring sexually transmitted infections, as well as mpox. The fact that 42 patients had sexual intercourse with confirmed mpox patients indicates the sexual transmissibility of this infection, further corroborating the transmission of the monkeypox virus during sexual intercourse. The attendance of parties or gatherings with multiple people could have contributed to the spread of the disease either by close skin-to-skin contact with an infected person or by facilitating the encounter of possible partners and subsequent engagement in unprotected sexual activities. While the specific circumstances of the remaining patients are unknown, it is important to consider the possibility of transmission through other means, such as close contact with infected individuals or exposure to contaminated surfaces. The provided epidemiological data highlight various potential risk factors and behaviors that could contribute to the transmission of infections among the patients. These include recent travel, engaging in unprotected sexual intercourse (both with unknown individuals and partners with confirmed mpox infection), and attending gatherings with possible opportunities for sexual activities. It is crucial to further investigate and analyze these findings to implement appropriate preventive measures and public health interventions to control the spread of infections.

Past medical history was notorious for concomitant HIV infection, which was found in 119 patients (corresponding to 40.9%) with confirmed mpox, with most patients having controlled disease. PrEP is a preventive measure for individuals at high risk of acquiring HIV, and 79 patients (27.1%) were actively taking this medication. These findings suggest that mpox possibly has risk factors common to HIV. Symptom presentation was relevant for fever (the most common symptom, affecting 60.8% of mpox patients), and localized mild-to-moderate pain, primarily in the anogenital area.

Cutaneous lesions were found to affect one to six body locations per patient, with the genital area being the most commonly involved, followed by the anal and perianal areas. The main affection of these sites indicates a strong association with sexual transmission during intercourse. This finding is particularly significant since mpox was traditionally considered to be primarily transmitted from animals to humans, but the current outbreak has demonstrated the virus’s ability to spread efficiently among humans through sexual intercourse. Additionally, the location of lesions in the anogenital region is characteristic of many infections that are transmitted sexually. Upper limbs, face, trunk, and oro-labial mucosa were also frequently affected. The cutaneous lesions were predominantly whitish papules or pseudo-pustules that became centrally umbilicated and covered in a hematic crust, eventually detaching spontaneously. Accompanying superficial lymphadenopathies were common, particularly in the inguinal region, reflecting the genital location of the skin lesions. These clinical data provide valuable insights into the medical conditions and symptomatology of the patients affected by mpox in this study.

The majority of patients were treated in ambulatory care and experienced a mild form of the disease, characterized by a small number of lesions and a good general state. No patient was treated with the aforementioned antiviral therapies. The precautionary measure of contact isolation was crucial for preventing the transmission of mpox to others. The incidence of complications was relatively low. Severe anal pain resembling acute proctitis was a complication found in nine patients, with resolution after oral and topical medication. Only three patients required hospital admission: one due to a bacterial superinfection of a mpox lesion on the face, and two for severe and refractory anal pain. These cases were successfully treated, resulting in a cure without any lasting effects or sequelae. No deaths directly attributed to mpox were registered among the patients in the study.

In this study, a comprehensive screening of 230 patients with confirmed mpox infection shed light on the prevalence of concomitant STIs in these individuals. About 46.5% of the confirmed mpox cases that undertook the STI screening had at least one concomitant STI, highlighting the significant association between these conditions. Among the observed concomitant STIs, *N. gonorrhoeae*, *C. trachomatis*, and syphilis were the most common. *N. gonorrhoeae* and *C. trachomatis* infections were detected in various locations, with the most common being the anorectal area in both agents. Furthermore, cases of early syphilis, primarily secondary or early latent syphilis, were identified in individuals with confirmed mpox. The presence of these bacterial STIs in conjunction with mpox suggests potential overlapping risk factors and the need for comprehensive sexual health screening and management during outbreaks.

Moreover, while relatively rare, inaugural infections with HIV and HCV were observed in a small number of patients with mpox. These viral infections are predominantly transmitted through sexual contact or exposure to infected blood [16,17]. The co-occurrence of HIV or HCV in individuals with mpox highlights the potential for multiple sexual health risks and the importance of comprehensive prevention strategies and education.

The previous findings lead to the conclusion that mpox probably progressed from an animal-to-human and/or generic human-to-human transmission to a disease with important sexual transmission. In fact, studies have revealed that monkeypox virus can be detected in various bodily fluids, including urine, saliva, semen, feces, oropharynx, and rectum [10]. Additionally, the fact that the main locations of mpox lesions were at the genital and anal/perianal areas further suggest transmission during sexual contact. Also, we found that a significant number of patients stated that they had unprotected sexual intercourse with mpox-confirmed partners. The infection occurred in a population of sexually active patients, with a history of multiple sexual partners. These findings demonstrate the sexual transmission route of mpox in the current outbreak and, as such, a thorough screening for other STIs should be prompted in patients suspected of having this infection.

Efforts should focus on promoting safe sexual practices, increasing awareness about the risks of co-infections, and ensuring access to comprehensive sexual health services, including testing, treatment, and prevention measures. Further research is warranted to better understand the underlying mechanisms behind the association between mpox and STIs, as well as to explore potential implications for disease transmission dynamics and management strategies.

## 5. Conclusions

We conclude that mpox can be sexually transmissible and emphasize the need for a thorough epidemiological and medical history, as well as a concomitant complete laboratorial screening for other STIs, including *N. gonorrhoeae*, *C. trachomatis*, syphilis, HIV, and HCV, with prompt treatment when applicable.

## Data Availability

Data are available on request due to restrictions (privacy). The data presented in this study are available on request from the corresponding author.

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
