# Peer review of "Demographic and Clinical Characteristics of Mpox Patients Attending an STD Clinic in Lisbon"

_ijerph, 2023, doi:10.3390/ijerph20196803_

Round 1

Reviewer 1 Report

The authors here present a case report of a cohort of 291 Mpox patients who presented to there hospital in 2022. They describe their demographics and clinical presentation of the infection, additionally, they also describe how a large number of the cohort also had concomitant sexually transmitted disease. The manuscript is well structured, concise, and easy to follow. Only minor linguistic errors were found, and would need to be ironed out. I find their findings interesting, given that few studies were able associate Mpox with other STIs, and this should certainly be kept in mind when treating patients with Mpox. However, I have some comments and questions that I hope the authors can address.

- Line 130: with only one case identified as female.

- Line 156 “Cutaneous lesions affected and average of 1.61 body locations” I understand that the number was is a statistically calculated one, but in a clinical context, it doesn’t make sense, please round up the numbers.

- Sub-section 3.5 belongs to the methodology section.

- In the conclusion section, while Mpox can indeed be transmitted through sexual contact, considering it as an STD requires rigorous testing and analysis, to establish more evidence. Given the lack of enough data to support this, I suggest the authors re-write this conclusion. By the way, the same is true for HCV, which is not really considered a STI, given the low level of transmission of this infection during sexual contact.

- Ethical statement: It is very unusual that the ethical permission was waived for this study. To my knowledge, for research limited to patients' medical records, access must be approved or cleared by an ethical review committee. In some countries retrospective analysis of routine clinical data is exempt from ethics approval requirements. However, this exemption normally only applies if the researchers have not had access to identifying patient information, which is not the case here. At some point someone must have had access to medical records. Please clarify this point.

Minor corrections for English language

Author Response

Dear Reviewer,

Thank you for your valuable suggestions, which have significantly improved the quality of our work. 

I've uploaded the new manuscript with track change.

Please find below the point-to-point answers to your revisions: 

- Line 130: with only one case identified as female. - This has been addressed.

- Line 156 “Cutaneous lesions affected and average of 1.61 body locations” I understand that the number was is a statistically calculated one, but in a clinical context, it doesn’t make sense, please round up the numbers. - This has been addressed.

- Sub-section 3.5 belongs to the methodology section. - This has been addressed.

- In the conclusion section, while Mpox can indeed be transmitted through sexual contact, considering it as an STD requires rigorous testing and analysis, to establish more evidence. Given the lack of enough data to support this, I suggest the authors re-write this conclusion. By the way, the same is true for HCV, which is not really considered a STI, given the low level of transmission of this infection during sexual contact. - This has been addressed.

- Ethical statement: It is very unusual that the ethical permission was waived for this study. To my knowledge, for research limited to patients' medical records, access must be approved or cleared by an ethical review committee. In some countries retrospective analysis of routine clinical data is exempt from ethics approval requirements. However, this exemption normally only applies if the researchers have not had access to identifying patient information, which is not the case here. At some point someone must have had access to medical records. Please clarify this point. - Ethical review and approval were waived for this article because this is a retrospective study and only de-identified patient information was utilized. Moreover, it is noteworthy to mention that a previously published article by our team (doi: 10.1111/jdv.18679) has already presented a preliminary segment of this dataset, and no ethical committee statement was required. This underscores the rationale behind the waiver of patient consent for our current study, which aligns with established ethical guidelines surrounding the use of anonymized data in research endeavors.

Once again, we sincerely appreciate your time and advice.

Best regards,

Margarida Caldeira

Reviewer 2 Report

Dear authors,

Here are suggestions and comments:

- Line 20: Replace "work" with "study".

- In keywords, avoid words that are in the title. Replace them.

- Line 51-53: The reference on concomitant infections is missing.

- Line 54-90: These two paragraphs can be summarized. Honestly, there is too much detailed information on clinical signs here, when the focus of the introduction would be better if it continued the pace of the previous paragraphs (public health, outbreak, etc.). However, a paragraph is missing that talks about the spread of the virus in Europe.

- Line 100: Replace "our Hospital Center" with the name of the Hospital.

- Line 107: Replace "our Hospital Center" with the name of the Hospital.

- Has the study been submitted to an ethics committee? Despite the confidentiality report, it is necessary that the study that uses human data be monitored by a research ethics committee.

- Item 3.2 has a terrible presentation.

- Why were no statistics applied to the study? Not even percentages with confidence intervals were calculated (about table 1 results).

- Line 268: Replace "Neisseria" with "N.".

- Line 264-273: This paragraph describes the results, but does not dialogue with the literature, nor does it attempt to explain the reasons for these findings. Explore.

- Line 275-279: Reference is missing.

- In general, the discussion needs to try to find explanations for the results obtained and make a real dialogue with other publications. This is missing from the text.

I congratulate the study, and I hope the authors understand that I try to bring the best possible version of the study that has such relevant data.

Kind regards ever.

Author Response

Dear Reviewer,

Thank you for your valuable suggestions, which have significantly improved the quality of our work. I've uploaded the new manuscript with track change.

Please find below the point-to-point answers to your revisions: 

  • Line 20: Replace "work" with "study". This has been addressed.
  • In keywords, avoid words that are in the title. Replace them. This has been addressed. New keywords were added.

  • Line 51-53: The reference on concomitant infections is missing. This has been addressed.

  • Line 54-90: These two paragraphs can be summarized. Honestly, there is too much detailed information on clinical signs here, when the focus of the introduction would be better if it continued the pace of the previous paragraphs (public health, outbreak, etc.). However, a paragraph is missing that talks about the spread of the virus in Europe. We added a new paragraph addressing the spread of the virus in Europe. We chose to keep the information regarding clinical signs in order to keep the minimum required words. 

  • Line 100: Replace "our Hospital Center" with the name of the Hospital. This has been addressed.

  • Line 107: Replace "our Hospital Center" with the name of the Hospital. This has been addressed.

  • Has the study been submitted to an ethics committee? Despite the confidentiality report, it is necessary that the study that uses human data be monitored by a research ethics committee. Ethical review and approval were waived for this article because this is a retrospective study and only de-identified patient information was utilized. Moreover, it is noteworthy to mention that a previously published article by our team (doi: 10.1111/jdv.18679) has already presented a preliminary segment of this dataset, and no ethical committee statement was required. This underscores the rationale behind the waiver of patient consent for our current study, which aligns with established ethical guidelines surrounding the use of anonymized data in research endeavors.

  • Item 3.2 has a terrible presentation. This has been addressed.

  • Why were no statistics applied to the study? Not even percentages with confidence intervals were calculated (about table 1 results). This has been addressed - percentages were added.

  • Line 268: Replace "Neisseria" with "N.". This has been addressed.

  • Line 264-273: This paragraph describes the results, but does not dialogue with the literature, nor does it attempt to explain the reasons for these findings. Explore. We have further explored this matter on the new manuscript. 

  • Line 275-279: Reference is missing. This has been addressed.

  • In general, the discussion needs to try to find explanations for the results obtained and make a real dialogue with other publications. This is missing from the text. We tried to explore this topic in the new manuscript.

Once again, we sincerely appreciate your time and advice.

Best regards,

Margarida Caldeira

Round 2

Reviewer 2 Report

Dear authors,

The improvements are satisfactory. Congratulations and wish you success.

Kind Regards